# A contrastive adversarial encoder for multi-omics data integration

**Ma Yinghua**[1], **Ahmad Khan**[1]*, **Yang Heng**[1], **Fiaz Gul Khan**[1],
**Afnan Aldhahri**[2]*, **Iftikhar Ahmed Khan**[3]

**1** Department of Computer Science, COMSATS University Islamabad Abbottabad Campus, Abbottabad,
Pakistan, **2** Department of Software Engineering, College of Computing, Umm Al-Qura University,
Makkah, Saudi Arabia, **3** Department of Computer Science and Information Technology, The University of
Lahore, Lahore, Pakistan

\* ahmadkhan@cuiatd.edu.pk (AK); amdhari@uqu.edu.sa (AA)

## Abstract

Early and accurate cancer detection is crucial for effective treatment, prognosis, and the advancement of precision medicine. Analyzing omics data is vital in cancer research. While using a single type of omics data provides a limited perspective, integrating multiple omics modalities allows for a more comprehensive understanding of cancer. Current deep models struggle to achieve efficient dimensionality reduction while preserving global information and integrating multi-omics data. This often results in feature redundancy or information loss, overlooking the synergies among different modalities. This paper proposes a contrastive adversarial encoder (CAEncoder) for multi-omics data integration to address this challenge. The proposed model combines a Vision Transformer (ViT) and a CycleGAN, trained in an end-to-end contrastive manner. The ViT is the encoder, utilizing self-attention, while the CycleGAN employs adversarial learning to ensure more discriminative and invariant latent space embeddings. Contrastive adversarial training improves representation quality by preventing information loss, eliminating redundancy, and capturing the synergies among different omics modalities. To ensure contrastive adversarial training, a composite loss function is used, consisting of a weighted combination of Adversarial Loss (Hinge Loss), Cycle Consistency Loss, and Triplet Margin Loss. The Adversarial Loss and Cycle Consistency Loss provide feedback from the CycleGAN, ensuring effective adversarial learning. Meanwhile, the Triplet Margin Loss promotes contrastive learning by pulling similar samples together and pushing dissimilar samples apart in the latent space. The performance of the CAEncoder is evaluated on downstream classification tasks, including both binary and multi-class classifications of five different cancer types. The results show that the model achieved a classification accuracy of up to 93.33% and an F1 score of 92.81%, outperforming existing advanced models. These findings demonstrate the potential of our method to enhance precision medicine for cancer through improved multi-omics data integration.

York at Oswego, UNITED STATES OF AMERICA

**Peer Review History:** PLOS recognizes the
benefits of transparency in the peer review
process; therefore, we enable the publication of
all of the content of peer review and author
responses alongside final, published articles.
The editorial history of this article is available
here: https://doi.org/10.1371/journal.pone.
0333134

**Data availability statement:** Our data can be
downloaded from: https://figshare.com/articles/
dataset/Figs1_png/19248078;
https://github.com/Yaolab-fantastic/MOCAT;
https://github.com/lanbiolab/DeepKEGG.

**Funding:** This research work was funded by Umm Al-Qura University, Saudi Arabia, under grant number: 25UQU4310136GSSR04.

**Competing interests:** The authors have declared that no competing interests exist.

## 1 Introduction

Cancer continues to be one of the most critical health challenges globally. In 2022, nearly 20 million new cancer cases were reported worldwide, resulting in 9.7 million deaths related to the disease. It is estimated that approximately one in five people will be diagnosed with cancer at some point in their lives, with one in nine men and one in twelve women expected to die from it [1]. Early cancer detection has traditionally relied on conventional machine learning algorithms and single-omics data [2–4]. However, single-omics data often falls short of capturing essential information from various biological layers. In contrast, integrating multi-omics data with deep learning has shown significant improvements over single-omics modalities [5–11].

The current research focuses on integrating various omics modalities to extract combined information for a more effective analysis of this critical disease. Wang et al. [12] utilize a transformer with multi-head self-attention and graph convolutional networks (GCN) to integrate these multi-omics modalities. Their results indicate an accuracy of 83.0% for Alzheimer's classification and 86.7% for breast cancer classification. Lan et al. [13] proposed an integration model called DeepKEGG, which leverages biological hierarchical modules in the local connections of nodes to improve interpretability. This model also includes a pathway self-attention mechanism to explore correlations between different samples. Additionally, Zheng et al. [14] introduced a method called GCFANet, which processes multimodal omics data through global and cross-modal feature aggregation, feature confidence learning, and a GCN branch. Experimental results demonstrate that this method effectively enhances the classification performance of multi-omics data [15]. Furthermore, Li et al. [16] introduced a novel end-to-end multi-omics Graph Neural Network (GNN) framework for cancer classification, utilizing heterogeneous multilayer graphs to integrate both intra-omics and inter-omics connections. For breast cancer subset classification, Huang et al. [17] proposed a deep-learning framework called DSCCN. This method conducts differential analysis on multi-omics expression data to identify differentially expressed genes and employs sparse canonical correlation analysis to extract highly correlated features among these genes. These features are then trained separately using a multi-task deep learning neural network to predict breast cancer subtypes.

Zhu et al. [18] proposed a supervised deep learning method called the Geometric Graph Neural Network (GGNN). This approach integrates genomic geometric features and protein interaction pathway information into the deep learning model. The Denoised Multi-Omics Integration Framework [19] consists of two key components: a distribution-based feature denoising algorithm (FSD)–aimed at reducing data dimensionality, and a multi-omics integration framework (AttentionMOI)–designed for predicting cancer prognosis and identifying cancer subtypes. The results demonstrated that this model performed significantly well across 15 cancers in the TCGA database. The moBRCA-net framework [20] addresses the challenge of high-dimensional data in breast cancer classification. By integrating multi-omics data, it utilizes a feature selection module and a self-attention module to capture the relative importance of each omics modality. Deep Centroid [21] addresses challenges in omics data classification, including high-dimensional data, limited sample sizes, and source bias. Yan et al. [22] developed a hierarchical multi-level Graph Neural Network (GNN) approach that utilizes multi-omics data, gene regulatory networks, and pathway information to extract discriminative features, thereby improving the accuracy of survival risk predictions. AUTOSurv [23] utilizes a specially designed Variational Autoencoder (VAE) for the dimensionality reduction of multi-omics data. This model has demonstrated significant performance in prognosis prediction across multiple independent datasets when compared to alternative strategies and

machine learning methods. Guo et al. [24] utilize network embedding technology to integrate gene co-expression data, somatic mutation data, and clinical information. By combining the struc2vec model with the random survival forest (RSF) model, they successfully predicted both long-term and short-term survival outcomes for patients with lung adenocarcinoma (LUAD).

Multi-omics data integration models have demonstrated significant improvements in cancer analysis compared to single-omics models. However, these multi-omics models still face challenges in effectively capturing synergistic features from different modalities. This limitation undermines the full potential of data integration in cancer research. Additionally, multi-omics models often prioritize stronger modalities at the expense of weaker ones, which diminishes the benefits of joint learning and negatively impacts performance in downstream tasks. Furthermore, the imbalanced nature of the data affects the overall efficiency of these models.

This paper presents a novel multi-omics integration model for cancer classification. The framework includes two main components: 1) an encoder, which utilizes a transformer to map multi-omics data to a reduced latent space, and 2) a CycleGAN that provides feedback to the encoder, enabling it to learn discriminative features and enhance generalization. The model is trained in a supervised contrastive manner, which helps bring similar modalities closer together while distancing dissimilar ones in the latent space. By employing contrastive learning, the model manages to relatively effectively mitigate the data imbalance, ensuring that all modalities are taken into account and thereby learning the synergies across them. Finally, the classification is performed in the latent space. The results indicate a significant improvement compared to current state-of-the-art methods.

## 2 Proposed model

The proposed model (see Fig 1) comprises two main modules: the encoder and the Cycle-GAN [25]. The encoder is the vision transformer (ViT) model [26], denoted as $E : x \rightarrow z$, which maps the high-dimensional multi-omics modalities $x \in \mathbb{R}^{n_x}$ into a reduced latent space $z \in \mathbb{R}^{n_z}$, where $n_x$ is significantly greater than $n_z$. At the same time, the CycleGAN enhances the performance of the transformer encoder by providing feedback to integrate multi-omics information, extract discriminative features, and reduce dimensionality.

Both modules are trained in an end-to-end manner, where the CycleGAN provides gradient-based feedback to the ViT encoder via adversarial and cycle-consistency losses. This joint optimization enables the encoder to learn not only from contrastive objectives but also from the reconstruction feedback, which enhances its robustness and generalization. During inference, only the ViT encoder is used to extract low-dimensional representations for downstream classification tasks.

### 2.1 ViT encoder

The $E : x \rightarrow z$ is composed of 8 blocks, each consisting of multi-head attention (MHA), Switch Normalization (SN), and a Multi-layer Perceptron (MLP). The multi-head self-attention projects multi-omics data into subspaces calculates attention weights based on the significance of each position, and aggregates the outputs to produce the final attention output. The attention weight $\alpha_{ijk}$ for a position $j$ relative to all positions $k$ in the $i$th head is computed as:

$$\alpha_{ijk} = \text{softmax}\left(\frac{Q_i K_i^T}{\sqrt{d_k}}\right) \tag{1}$$

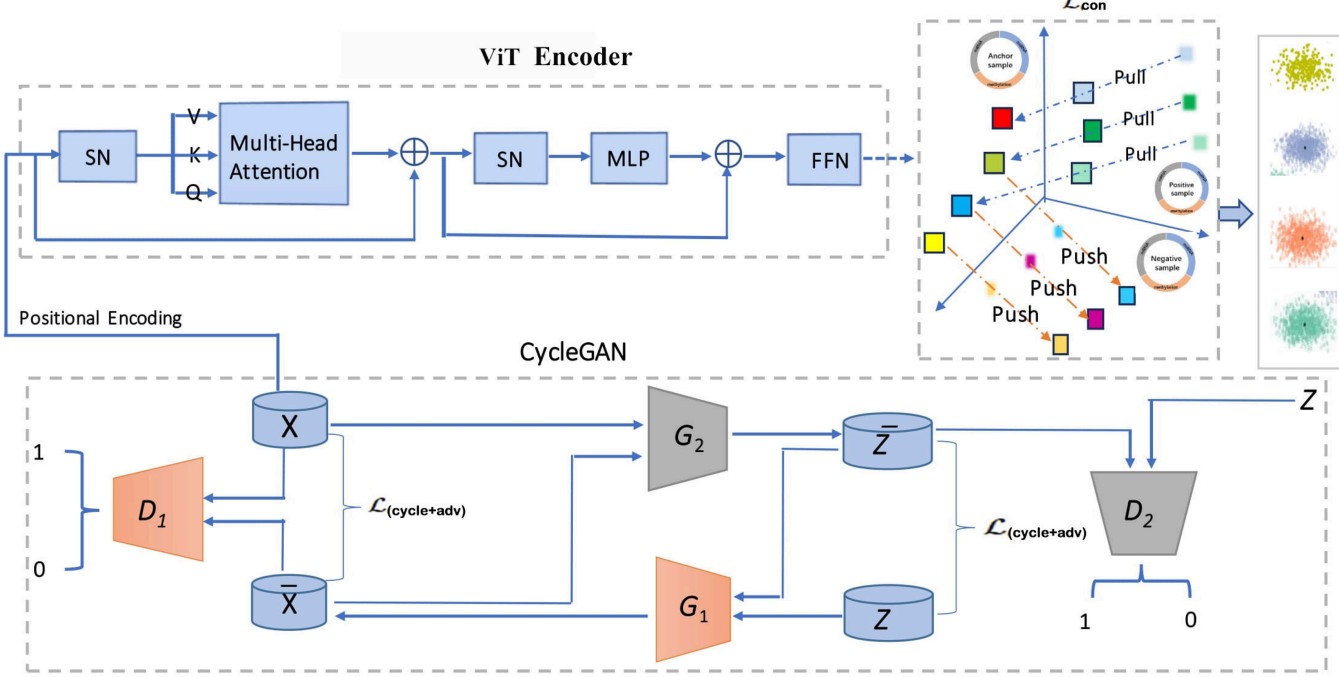

**Fig 1. The block diagram of the proposed Model.** Here, $X$ and $\bar{X}$ represent the original and synthesized multi-omics data, while $Z$ and $\bar{Z}$ denote the latent reduced space. The acronyms *SN*, *MLP*, *ViT*, and *FFN* stand for Switch Normalization, Multi-Layer Perceptron, Vision Transformer, and Feedforward Network, respectively. The symbols $D_1$ and $D_2$ refer to Discriminator 1 and Discriminator 2, while $G_1$ and $G_2$ represent Generator 1 and Generator 2. The symbol $\oplus$ indicates component-wise addition.

where $Q_i = X_i W_i^Q$, $K_i = X_i W_i^K$, $X_i$ is the multi-omics modality, $W_i^Q$ is the query weight matrix, and $W_i^K$ is the key weight matrix. So, the output of the $i$th attention head is given by:

$$O_i = \sum_j \alpha_{ijk} V_i \tag{2}$$

where $V_i = X_i W_i^V$ is the value matrix. Finally, the outputs from all attention heads are concatenated, $O = [O_1, O_2, \dots, O_h]$, to create the final multi-head attention output.

The normalized output, denoted as $O \in \mathbb{R}^{1024}$, from the final multi-head attention layer is passed through a feedforward network (FFN) that consists of three layers. This network progressively transforms the output into a reduced latent vector $z \in \mathbb{R}^{256}$. The first two layers of the FFN utilize the Leaky ReLU activation function, while the final layer is linear. Typically, transformers use a class token, denoted as *CLS*, for classification purposes; however, in this case, the *CLS* token is not employed because the goal is to map the high-dimensional data into a reduced latent space.

Here, the Switch Normalization (SN) [27] is used to improve both the training stability and expressive capability of the encoder. Unlike fixed normalization techniques, SN dynamically selects the most suitable normalization strategy based on the input characteristics and training conditions. The trainable coefficients are used to combine Batch Normalization (BN) and Layer Normalization (LN) to ensure Switch Normalization (SN). Given an input feature $x$, the SN output is computed as:

$$SN(x) = \alpha \cdot BN(x) + \beta \cdot LN(x), \tag{3}$$

where $\alpha$ and $\beta$ are trainable parameters that adaptively balance the contributions of BN and LN during training. This dynamic adjustment allows SN to optimize the model's adaptability to varying data distributions, enhancing training stability and generalization performance. The experiments demonstrate that SN effectively stabilizes training by mitigating the sensitivity to batch size variations and distribution shifts.

Moreover, although the primary objective of the ViT encoder is driven by contrastive loss, its parameters are also influenced by the adversarial and cycle-consistency losses propagated from the CycleGAN. Since both generators $G_1$ and $G_2$ operate on the latent representation $z = E(x)$, the feedback from their respective losses flows back to the encoder. This integrated training mechanism allows the ViT encoder to refine its feature extraction by leveraging both discriminative and generative signals.

Overall, the encoder $E$ is composed of 8 Transformer blocks, each with 8 attention heads. To project the high-dimensional input into a representation suitable for Transformer encoding, a linear projection layer is first applied to map the input to a 1024-dimensional embedding space. Following the attention mechanism, a multi-layer perceptron (MLP) consisting of three fully connected layers is used, with output dimensions of $1024 \rightarrow 512 \rightarrow 256$. The first two layers employ the Leaky ReLU activation function, and the final layer is linear.

## 2.2 CycleGAN architecture

The proposed CycleGAN architecture consists of two generators, $G_1$ and $G_2$, and two discriminators, $D_1$ and $D_2$. This framework enables bidirectional translation between high-dimensional multi-omics data and their corresponding low-dimensional latent representations.

- Generator $G_1 : z \rightarrow \bar{x}$ takes a latent representation $z$, obtained from the encoder $E$, and generates a synthetic multi-omics modality $\bar{x}$ that approximates the original high-dimensional data $x$.
- Generator $G_2 : x \rightarrow \bar{z}$ takes the original multi-omics data $x$ as input and reconstructs a latent representation $\bar{z}$ that should resemble the true latent vector $z = E(x)$.
- Discriminator $D_1 : \{x, \bar{x}\} \rightarrow \{1, 0\}$ attempts to distinguish real high-dimensional data $x$ from the generated data $\bar{x}$.
- Discriminator $D_2 : \{z, \bar{z}\} \rightarrow \{1, 0\}$ aims to differentiate between the true latent vector $z$ and the reconstructed vector $\bar{z}$.

The objectives are as follows:

- Generator $G_1$ is trained to ensure that the generated data $\bar{x} = G_1(z)$ is indistinguishable from the real multi-omics data $x$.
- Generator $G_2$ is trained to ensure that the reconstructed latent representation $\bar{z} = G_2(x)$ closely approximates the true latent vector $z = E(x)$.
- Discriminator $D_1$ is trained to assign a high score to real samples and a low score to generated ones:

$$D_1(x) \approx 1, \quad D_1(\bar{x}) \approx 0 \tag{4}$$

- Discriminator $D_2$ is trained similarly to distinguish real and generated latent vectors:

$$D_2(z) \approx 1, \quad D_2(\bar{z}) \approx 0 \tag{5}$$

As shown in Eqs 4 and 5, the discriminators aim to maximize the prediction scores for real samples while minimizing them for generated ones, thereby guiding the generators to produce realistic outputs.

Originally, CycleGAN was developed for image generation using 2D convolution. However, in our case, the input multi-omics data $x \in \mathbb{R}^{n_x}$ is a long vector. Therefore, we have adapted CycleGAN to utilize linear convolution.

1D convolutions operate along the feature vector, enabling the model to capture dependencies across different omics features. The 1D convolution operation can be expressed as:

$$y_i = \sum_{j=0}^{k-1} w_j \cdot x_{i+j} + b \tag{6}$$

where $x \in \mathbb{R}^{n_x}$ is the input feature vector, $w \in \mathbb{R}^k$ is the convolution kernel, $b$ is the bias term, and $k$ is the kernel size. Furthermore, the generators $G_1$ and $G_2$ are designed to map input feature vectors into output vectors using a combination of linear layers and 1D convolutions. The discriminators $D_1$ and $D_2$ also operate on feature vectors, ensuring that adversarial training remains robust and effective for structured data. These modifications allow CycleGAN to model bidirectional mappings between different multi-omics modalities while preserving the advantages of adversarial training.

Since all generated and reconstructed data in CycleGAN rely on the latent vectors $z$ produced by the encoder $E$, the encoder is directly updated through the gradients of adversarial and cycle-consistency losses. This design effectively couples CycleGAN with the encoder, enhancing the encoder's feature learning capability beyond what contrastive learning alone can provide.

The entire model, integrating the ViT encoder with CycleGAN, is optimized using a supervised contrastive learning approach. This contrastive mechanism enables us to bring similar points closer together in the latent space while pushing dissimilar points further apart. Additionally, it facilitates the understanding of synergies among different modalities, ultimately enhancing the performance of downstream classification tasks.

After transforming the high-dimensional multi-omics data $x \in \mathbb{R}^{n_x}$ into the corresponding latent space $z \in \mathbb{R}^{n_z}$, the classification task is then conducted within the latent space $z \in \mathbb{R}^{n_z}$.

## 2.3 Loss functions

The model is trained end-to-end by integrating three types of losses: contrastive loss, adversarial loss, and cycle consistency loss. The objective is to train the encoder $E(\cdot)$ using a contrastive adversarial approach, effectively mapping high-dimensional data into a compact latent space. This process ultimately enhances the performance of downstream classification tasks.

**Contrastive Loss:** The contrastive loss [28] optimizes the encoder $E$ to reduce the distance between similar samples while increasing the distance between dissimilar samples within the latent space $z$.

$$\mathcal{L}_{\text{con}} = \sum_{i=1}^{N} \Big( \left\| E(x_i^a) - E(x_i^p) \right\|_2^2 \\ + \max\left(0, m - \left\| E(x_i^a) - E(x_i^n) \right\|_2^2\right) \Big) \tag{7}$$

where $E(\cdot)$ represents the encoder, $x_i^a$ represents anchor samples, $x_i^p$ represents positive samples, $x_i^n$ represents negative samples, $N$ is the batch size, and $m$ is the margin used in the contrastive loss.

**Adversarial loss**: The Hinge Loss [27] is used as an adversarial loss because it offers better stability and faster convergence compared to traditional cross-entropy loss. This approach is particularly effective for handling high-dimensional data, as it more effectively manages the adversarial dynamics between the generator and discriminator.

The generator $G_1$ produces synthetic high-dimensional multi-omics data $\bar{x}$ from low-dimensional latent vectors $z$ and $\bar{z}$. Meanwhile, the discriminator $D_1$ tries to differentiate between real high-dimensional multi-omics data $x$ and the synthetic multi-omics data $\bar{x}$. Consequently, the adversarial loss can be defined as follows:

$$
\begin{aligned}
\mathcal{L}_{G1_{\mathrm{adv}}} = -\frac{1}{3} \sum_{i=1}^{N} \Bigg( & \mathbb{E}_{z^a \sim p_Z}\big[\min\big(0, -1 + D_1\big(G_1(z^a)\big)\big)\big] \\
& + \mathbb{E}_{z^p \sim p_Z}\big[\min\big(0, -1 + D_1\big(G_1(z^p)\big)\big)\big] \\
& + \mathbb{E}_{z^n \sim p_Z}\big[\min\big(0, -1 + D_1\big(G_1(z^n)\big)\big)\big] \Bigg)
\end{aligned}
\tag{8}
$$

In a similar manner, the generator $G_2$ transforms high-dimensional multi-omics data $x$ into a low-dimensional latent vector $\bar{z}$, while the discriminator $D_2$ aims to differentiate between $z$ and $\bar{z}$.

$$
\begin{aligned}
\mathcal{L}_{G2_{\mathrm{adv}}} = -\frac{1}{3} \sum_{i=1}^{N} \Bigg( & \mathbb{E}_{x^a \sim p_X}\big[\min\big(0, -1 + D_2\big(G_2(x^a)\big)\big)\big] \\
& + \mathbb{E}_{x^p \sim p_X}\big[\min\big(0, -1 + D_2\big(G_2(x^p)\big)\big)\big] \\
& + \mathbb{E}_{x^n \sim p_X}\big[\min\big(0, -1 + D_2\big(G_2(x^n)\big)\big)\big] \Bigg)
\end{aligned}
\tag{9}
$$

The objective of the discriminator $D_1$ is to minimize the output value for generated data, pushing it closer to $-1$. This process enables $D_1$ to effectively distinguish between real high-dimensional multi-omics data $x$ and the generated data $G(z)$.

$$
\begin{aligned}
\mathcal{L}_{D_1} = \frac{1}{6} \sum_{i=1}^{N} \Big( & \mathbb{E}_{x_i^a \sim p_X}\big[\max\big(0, 1 - D_1(x_i^a)\big)\big] \\
& + \mathbb{E}_{x_i^p \sim p_X}\big[\max\big(0, 1 - D_1(x_i^p)\big)\big] \\
& + \mathbb{E}_{x_i^n \sim p_X}\big[\max\big(0, 1 - D_1(x_i^n)\big)\big] \\
& + \mathbb{E}_{z_i^a \sim p_Z}\big[\max\big(0, 1 + D_1\big(G_1(z_i^a)\big)\big)\big] \\
& + \mathbb{E}_{z_i^p \sim p_Z}\big[\max\big(0, 1 + D_1\big(G_1(z_i^p)\big)\big)\big] \\
& + \mathbb{E}_{z_i^n \sim p_Z}\big[\max\big(0, 1 + D_1\big(G_1(z_i^n)\big)\big)\big] \Big)
\end{aligned}
\tag{10}
$$

The discriminator $D_2$ aims to minimize the output value of the generated data, aiming to make it as close to $-1$ as possible. This process effectively differentiates between $z = E(x)$ and $\bar{z} = G_2(x)$.

$$\mathcal{L}_{D_2} = \frac{1}{6} \sum_{i=1}^{N} \Big( \mathbb{E}_{z_i^a \sim p_Z}[\max(0, 1 - D_2(z_i^a))]$$
$$+ \mathbb{E}_{z_i^p \sim p_Z}[\max(0, 1 - D_2(z_i^p))]$$
$$+ \mathbb{E}_{z_i^n \sim p_Z}[\max(0, 1 - D_2(z_i^n))]$$
$$+ \mathbb{E}_{x_i^a \sim p_X}[\max(0, 1 + D_2(G_2(x_i^a)))]$$
$$+ \mathbb{E}_{x_i^p \sim p_X}[\max(0, 1 + D_2(G_2(x_i^p)))]$$
$$+ \mathbb{E}_{x_i^n \sim p_X}[\max(0, 1 + D_2(G_2(x_i^n)))] \Big) \tag{11}$$

where, $x_i^a$, $x_i^p$, and $x_i^n$ represent the anchor, positive, and negative samples, respectively, for high-dimensional multi-omics data. Similarly, $z_i^a$, $z_i^p$, and $z_i^n$ denote the latent low-dimensional vectors corresponding to these samples. The symbols $p_X$ and $p_Z$ refer to the real distributions of high-dimensional and low-dimensional data, respectively. The notation $\mathbb{E}_{x \sim p_x}[\cdot]$ and $\mathbb{E}_{z \sim p_Z}[\cdot]$ indicates the expectations over high-dimensional and low-dimensional data, respectively. Lastly, $\min(0, \cdot)$ represents the standard form used in Hinge Loss, ensuring that the generator's output is as close to 1 as possible, while the discriminator's output is as close to -1 as possible.

**Cycle Consistency Loss:** The cycle consistency loss is employed to ensure that the generator's output can be accurately mapped back to the original input, thereby maintaining data consistency. This is especially crucial when working with high-dimensional multi-omics data, as it helps preserve the complex structure and biological significance of the data, preventing the generated high-dimensional output from losing its original characteristics. Furthermore, cycle consistency loss indirectly enhances the feature extraction capability and training stability of the Transformer model by minimizing feature loss and ensuring data coherence. The cycle consistency loss for the generator $G_1$ is defined as follows:

$$\mathcal{L}_{G1_{\text{cycle}}} = \frac{1}{3} \sum_{i=1}^{N} \Big( \|G_2(G_1(z_i^a)) - z_i^a\|_1 + \|G_2(G_1(z_i^p)) - z_i^p\|_1 + \|G_2(G_1(z_i^n)) - z_i^n\|_1 \Big) \tag{12}$$

The cycle consistency loss for generator $G_2$ is given by:

$$\mathcal{L}_{G2_{\text{cycle}}} = \frac{1}{3} \sum_{i=1}^{N} \Big( \|G_1(G_2(X_i^a)) - X_i^a\|_1 + \|G_1(G_2(X_i^p)) - X_i^p\|_1 + \|G_1(G_2(X_i^n)) - X_i^n\|_1 \Big) \tag{13}$$

where the $L_1$ norm, denoted as $\|\cdot\|_1$, is used to compute the absolute error between the generated data and the original data.

**Total Loss:** The total loss is the weighted combination of the contrastive loss, $\mathcal{L}_{\text{con}}$, the cycle loss, $\mathcal{L}_{G1_{\text{cycle}}}$, and $\mathcal{L}_{G2_{\text{cycle}}}$; as well as the adversarial losses $\mathcal{L}_{G1_{\text{adv}}}$ and $\mathcal{L}_{G2_{\text{adv}}}$.

$$\mathcal{L}_{\text{total}} = \mathcal{L}_{\text{con}} + \alpha \left( \mathcal{L}_{G1_{\text{cycle}}} + \mathcal{L}_{G2_{\text{cycle}}} \right)$$
$$+ \beta \left( \mathcal{L}_{G1_{\text{adv}}} + \mathcal{L}_{G2_{\text{adv}}} \right) \tag{14}$$

where $\alpha$ and $\beta$ are weights that are used to balance the various losses. Therefore, the weights of the encoder are adjusted based on the total loss, $\frac{\partial \mathcal{L}_{\text{total}}}{\partial W}$, where $W$ represents the trainable parameters of the encoder $E$. Contrastive learning improves the distinction between positive

and negative samples, while the adversarial loss and cycle consistency loss of GAN provide feedback to the encoder $E$, enhancing feature extraction and resulting in better generalization.

In this integrated setup, the ViT encoder benefits not only from contrastive discrimination but also from reconstruction-based supervision, as the gradients from both the generators and discriminators in CycleGAN are backpropagated through the encoder. This unified feedback loop improves both representation quality and model robustness.

## 3 Experiments

The section discusses the dataset, the training and hyper-parameters setting of the model, and the quantitative results in detail.

### 3.1 Datasets

To illustrate the effectiveness of the CAEncoder, we utilized three cancer datasets from TCGA [29] and ROSMAP. Dataset-1 [30] is sourced from the TCGA repository and is referred to as 4-BRCA. This dataset includes multi-omics data such as Copy Number Variation (CNV), mRNA, and Reverse Phase Protein Array (RPPA) data. It encompasses four subtypes of breast cancer: Basal-like, Her2-enriched, Luminal A, and Luminal B, with a total of 511 samples. Dataset-2 [13] combines Alzheimer's binary classification data from ROSMAP and BRCA five-class data from TCGA. It includes various modalities such as mRNA, DNA methylation, and miRNA. This dataset contains 169 samples from Alzheimer's disease (AD) patients and 182 samples from normal controls (NC), while the five-class BRCA dataset comprises 875 samples. Dataset-3 [31] consists of data from TCGA, covering four cancer types: Prostate Adenocarcinoma (PRAD) with 250 samples, Breast Invasive Carcinoma (BRCA) with 211 samples, Bladder Urothelial Carcinoma (BLCA) with 402 samples, and Liver Hepatocellular Carcinoma (LIHC) with 354 samples. It features three modalities: mRNA, Single Nucleotide Variants (SNV), and miRNA.

**Data Preprocessing:** To ensure the quality and consistency of multi-omics data, categorical variables in each omics type (e.g., copy number variation (CNV), mRNA, and reverse phase protein array (RPPA)) are converted into numerical variables. All features are normalized to have a mean of 0 and a standard deviation of 1, which helps maintain consistency across different omics datasets. Furthermore, we incorporate all features from each omics type into the model, allowing it to fully capture the complex biological relationships present in the multi-omics data. To address the inherent imbalance in multi-omics data across different cancer types, we constructed balanced sets of positive and negative sample pairs for each cancer type, with the proportions reflecting their respective sample sizes. The encoder is trained in a contrastive manner, aiming to group similar points closer together in the latent space while pushing dissimilar points farther apart. Contrastive learning requires the samples to be divided into three categories: anchor, positive, and negative.

**Anchor, positive and negative samples generation:** We consider a multi-omics dataset comprising $N$ samples, each containing data from $M$ distinct modalities. For a given modality $m$ (where $1 \leq m \leq M$), the samples can be described as $X_m = \{(x_m^i, y^i) | 1 \leq i \leq N\}$, where $x_m^i \in \mathbb{R}^{n_m}$ is a feature vector for the $i$-th sample and $y^i$ indicates its cancer subtype. An anchor sample is formed by combining feature vectors from $m$ selected modalities of the same patient and subtype: $x^a = [x_1^i; x_2^i; \ldots; x_m^i]$. For instance, for the $i$-th patient classified as *Luminal-A* breast cancer, $x_A = [x_{CNV}^i, x_{mRNA}^i, x_{RPPA}^i]$. A positive sample $x^p$ is created by selecting $m$ modalities from different patients who share the same subtype as the anchor. It is expressed as $x^p = [x_1^{k_1}; x_2^{k_2}; \ldots; x_m^{k_j}]$, where $k_j \sim U([1, N])$. For example, if patients $i$, $j$, and $k$ all have the *Luminal-A* subtype, the positive sample could be $x^p = [x_{CNV}^i, x_{mRNA}^j, x_{RPPA}^k]$. Conversely,

a negative sample $x^n$ is generated by selecting at least one modality from a patient with a different class label. It is similarly represented, but at least one of the labels $y^k$ must differ from that of the anchor. For instance, if $y^i \neq y^j$, then a possible negative sample could be $x^n = [x^i_{\text{CNV}}, x^i_{\text{mRNA}}, x^j_{\text{RPPA}}]$. Using $k \sim U([1, N])$ for sample selection introduces variability and adds complexity to the training, often resulting in positive samples that are more dissimilar to the anchor despite sharing the same label. This approach enhances the model's ability to discern subtle differences between cancer subtypes and improves robustness by exposing it to challenging examples, ultimately assisting in better generalization and reducing the risk of overfitting.

## 3.2 Training and hyper-parameters setting

To ensure reliable and statistically significant results, we adopted a 5-fold cross-validation protocol across all datasets. Each dataset was randomly split into five folds, with 80% used for training and 20% for testing in each iteration. This process was repeated five times using different random seeds to ensure generalizable performance. The encoder was trained using the Adam optimizer with a learning rate of $\alpha = 0.0001$. We applied gradient clipping (1.0) to prevent gradient explosion and incorporated L2 regularization (weight decay) to mitigate overfitting. The model was trained for 100 epochs with a batch size of 64.

Experiments were conducted on a system with an Intel Core i7-10700K CPU at 3.80GHz, paired with an NVIDIA GeForce RTX 3080 GPU. It has 32GB of DDR4 RAM and a 1TB SSD for fast data processing. The operating system is Ubuntu 20.04 LTS, and TensorFlow 2.5.0 was used for deep learning, with all scripts executed in Python 3 for compatibility with the latest libraries.

Our proposed model consists of a Transformer encoder and a CycleGAN framework. The computational complexity of the Transformer encoder follows Vaswani et al. [32] and can be expressed as:

$$O(L(n_t^2 d_t + n_t d_t^2)), \tag{15}$$

where $L$ is the number of layers, $n_t$ is the number of samples, and $d_t$ is the feature dimension.

The computational complexity of the CycleGAN framework can be analyzed based on the complexity of convolutional networks [25,33–35]. For a single convolutional layer, the complexity is given by:

$$O(2n_c \kappa d_c + 2n_c d_c^2), \tag{16}$$

where $n_c$ is the number of samples, $\kappa$ is the convolutional kernel size, and $d_c$ is the feature dimension.

Overall, the combined computational complexity remains manageable for our dataset, allowing for efficient model training within a reasonable time frame.

## 3.3 Results

**Classifier selection:** The CAEncoder maps high-dimensional multi-omics data into a reduced latent space, making it challenging to evaluate the effectiveness of the proposed encoder directly. Therefore, we assess its performance based on downstream task classification. To accomplish this, we experimented with various classifiers, including Random Forest (RF), k-nearest Neighbors (K-NN), Decision Tree (DT), and Gradient Boosting Classifier (GBC), which were trained in the reduced latent space.

Table 1 presents the classification results of various classifiers operating in the reduced latent space. The experiments indicate that the Random Forest (RF) classifier outperforms other classifiers in effectively utilizing this latent representation, achieving the highest performance. This effectiveness arises from its ensemble learning strategy, which combines predictions from multiple decision trees. This approach reduces model variance, enhances robustness to noise, and minimizes the risk of overfitting. Furthermore, RF trains multiple sub-models on different subsets of features, thereby leveraging complementary information from various modalities to boost classification performance. Consequently, we have chosen RF as the final classifier for our model in the subsequent experiments.

**The performance of CAEncoder across various modalities:** Fig 2 displays the performance of the CAEncoder model on Dataset-1, showcasing various combinations of modalities: one-modality, two-modalities, and three-modalities. The CAEncoder model first transforms the high-dimensional data into a latent space, after which classification is conducted using a Random Forest (RF) algorithm with 15 estimators ($n\_estimators = 15$). The figure illustrates that combining more modalities results in improved performance. This demonstrates that the proposed encoder effectively learns the synergies among different modalities, enhancing generalization.

**Table 1. The performance of various classifiers on Dataset-1.** Initially, the dataset is transformed into a reduced latent space $z$ using the proposed encoder $E$, after which classification is performed in this reduced space.

| Classifier | ACC | F1 |
|---|---|---|
| K-NN | 0.7184 ± 0.0129 | 0.6935 ± 0.0155 |
| DT | 0.8928 ± 0.0200 | 0.8581 ± 0.0191 |
| GBC | 0.8946 ± 0.0167 | 0.8903 ± 0.0172 |
| **RF** | **0.9333 ± 0.0188** | **0.9281 ± 0.0131** |

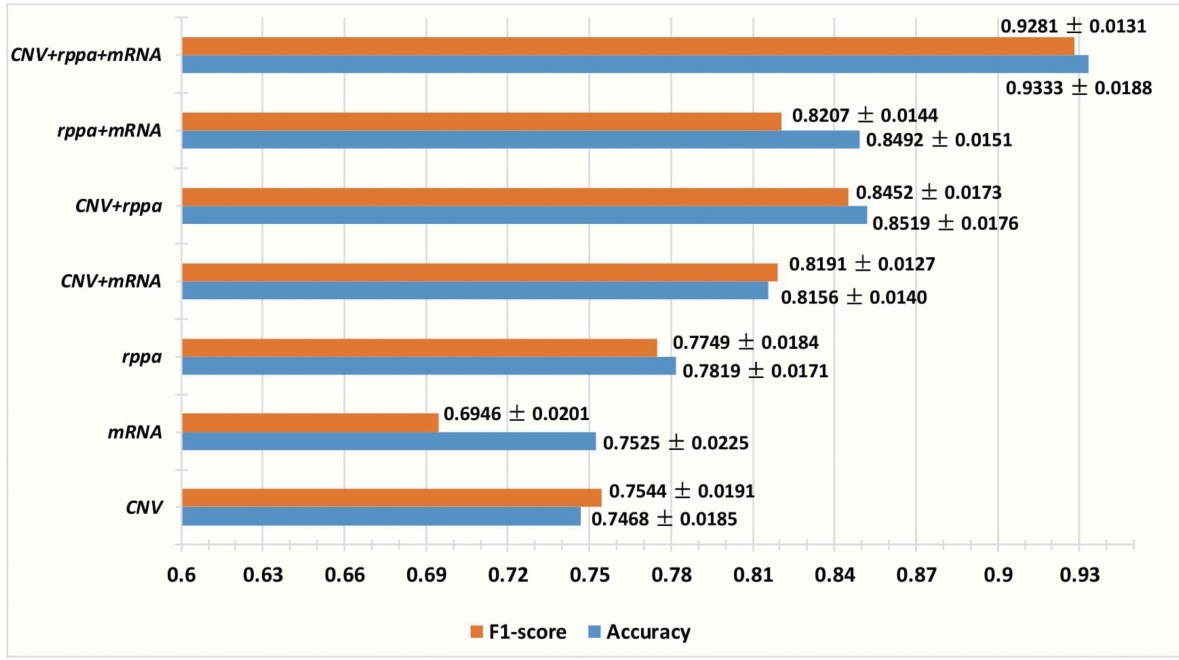

**Fig 2. Comparison of classification performance on various combinations of the multi-omics modalities.**

**Results on Dataset-1:** Table 2 presents a comparison of the classification performance between our proposed CAEncoder model and traditional machine learning classifiers, including Support Vector Machine (SVM), Multilayer Perceptron (MLP), and Convolutional Neural Network (CNN), using Dataset-1. In this study, we combined three modalities—CNV, RPPA, and mRNA—for classification purposes. To reduce dimensionality and extract independent features, we applied Principal Component Analysis (PCA) before training the aforementioned classifiers in the reduced feature space. We determine the number of retained principal components by examining the cumulative explained variance ratio, as outlined in [36]. We select components that account for at least 95% of the total variance. This threshold is chosen to balance dimensionality reduction with information retention. Our goal is to reduce the dimensionality of the data for improved computational efficiency while preserving the essential information needed to maintain model performance. The results indicate that the proposed CAEncoder model outperforms the traditional machine learning approaches.

Additionally, Table 2 compares the CAEncoder model with the Multi-Omics Integration Method Based on Graph Convolutional Network for Cancer Subtype Analysis (MoGCN) [30]. The MoGCN is a deep learning model that also integrates the modalities CNV, RPPA, and mRNA from Dataset-1 for classification. The results for MoGCN [30] are not recreated here but have been cited from the original paper. The findings demonstrate that the CAEncoder model outperforms MoGCN in terms of ACC and F1 scores by 3.51% and 2.65%, respectively.

**Results on Dataset-2:** In this section, we compare the classification performance of the proposed CAEncoder model with several state-of-the-art deep learning approaches, including MOGONET [37], MODILM [38], HyperTMO [5], and MOCAT [31]. MOGONET [37] integrates multi-omics data using graph convolutional networks, enabling patient classification and biomarker identification. MODILM [38] enhances classification accuracy for complex diseases by synthesizing significant and complementary information from various single-omics datasets. HyperTMO [5] is a multi-omics integration framework specifically designed for patient classification. It utilizes a hypergraph convolutional network to construct hypergraph structures that represent associations between samples in single-omics data. Evidence extraction is performed via the hypergraph convolutional network, allowing for the integration of multi-omics information at an evidence level. The Multi-Omics Integration Framework with Auxiliary Classifiers-enhanced Autoencoders (MOCAT) [31] effectively leverages intra- and inter-omics information. It employs attention mechanisms combined with confidence learning to improve feature representation and ensure trustworthy predictions. The results from these approaches are cited directly from their respective papers and have not been regenerated. Table 3 presents the results of the proposed method alongside the state-of-the-art methods on Dataset-2. The results indicate that the proposed CAEncoder outperforms all the SOTA methods in terms of accuracy and F1 scores.

**Table 2**. **Results on Dataset-1.**

|  | Acc | F1 |
|---|---|---|
| *SVM* | 0.8494 ± 0.0360 | 0.8447 ± 0.0351 |
| *MLP* | 0.8738 ± 0.0221 | 0.8772 ± 0.0268 |
| *CNN* | 0.8058 ± 0.0378 | 0.7852 ± 0.0311 |
| *MoGCN* [30] | 0.8982 ± 0.0246 | 0.9016 ± 0.0235 |
| *CAEncoder* | **0.9333 ± 0.0188** | **0.9281 ± 0.0131** |

**Table 3. Results on Dataset-2.**

| | ROSMAP (2 Categories) | | BRCA (5 Categories) | |
|---|---|---|---|---|
| | *ACC* | *F1* | *ACC* | *F1* |
| *MOGONET* [37] | 0.815 ± 0.023 | 0.821 ± 0.022 | 0.829 ± 0.018 | 0.825 ± 0.016 |
| *MODILM* [38] | 0.843 ± 0.000 | 0.850 ± 0.000 | 0.845 ± 0.000 | 0.840 ± 0.000 |
| *HyperTMO* [5] | 0.875 ± 0.033 | 0.874 ± 0.033 | 0.858 ± 0.023 | 0.863 ± 0.023 |
| *MOCAT* [31] | 87.6[*](86.7–88.5) | 87.5[*](86.8–88.2) | 88.5[*](88.1–88.9) | 88.9[*](88.5–89.3) |
| **CAEncoder** | **0.909 ± 0.024** | **0.894 ± 0.019** | **0.914 ± 0.034** | **0.914 ± 0.027** |

**Results on Dataset-3:** Table 4 compares the performance of CAEncoder and DeepKEGG [13] across all four cancer types included in Dataset-3. DeepKEGG is an interpretable multi-omics data integration method designed to predict cancer recurrence and identify biomarkers. It features a biological hierarchical module that establishes local connections between neuron nodes, enhancing the model's interpretability by illustrating the relationships among genes, miRNAs, and pathways. Additionally, it includes a pathway self-attention module, which analyzes the correlations between different samples and generates potential pathway feature representations that improve the model's prediction performance. The results indicate that CAEncoder outperforms DeepKEGG in all four areas.

## 3.4 Ablation study

The proposed model, CAEncoder, primarily consists of an encoder (a transformer) and CycleGAN, and it is trained using a contrastive approach. To assess the effectiveness of each component, we conducted ablation experiments (refer to Table 5).

In the first experiment, we kept both the encoder and the CycleGAN intact but bypassed the contrastive loss, resulting in a model variant we named CAEncoder_notCL. This experiment aimed to highlight the significance of contrastive learning. The results indicated that omitting contrastive learning significantly affected the model's performance.

In the second experiment, we removed the CycleGAN while retaining the other components, creating a model referred to as CAEncoder_RC. The results from this experiment demonstrated that each component of the proposed model plays a crucial role in enhancing classification performance.

**Table 4. Results on Dataset-3.**

| | DeepKEGG [13] | | CAEncoder | |
|---|---|---|---|---|
| | *ACC* | *F1* | *ACC* | *F1* |
| *BRCA* | 0.784 ± 0.006 | 0.672 ± 0.007 | 0.8602 ± 0.021 | 0.8511 ± 0.023 |
| *LIHC* | 0.877 ± 0.006 | 0.870 ± 0.007 | 0.8986 ± 0.015 | 0.9060 ± 0.021 |
| *PRAD* | 0.736 ± 0.007 | 0.679 ± 0.006 | 0.8908 ± 0.032 | 0.8753 ± 0.029 |
| *BLCA* | 0.896 ± 0.003 | 0.857 ± 0.006 | 0.9284 ± 0.014 | 0.9110 ± 0.016 |

**Table 5. Results of the ablation study using Dataset-1.**

| | Acc | F1 |
|---|---|---|
| *CAEncoder_notCL* | 0.8407 ± 0.0389 | 0.8428 ± 0.0378 |
| *CAEncoder_RC* | 0.8024 ± 0.0251 | 0.8053 ± 0.0276 |
| *CAEncoder* | **0.9333 ± 0.0188** | **0.9281 ± 0.0131** |

These ablation experiments provide compelling evidence of how the CAEncoder learns more discriminative and generalizable representations. Specifically, the substantial performance drop observed in CAEncoder_notCL confirms the pivotal role of contrastive learning in enhancing feature separability within the latent space. Similarly, the reduced accuracy and F1-score in CAEncoder_RC highlight the contribution of CycleGAN in preserving modality-specific details and preventing the loss of global structural information. Together, these components synergistically improve the quality of learned representations, leading to better generalization on downstream classification tasks.

## 4 Conclusion

This study introduces a novel multi-omics integration model called CAEncoder for cancer classification. This framework effectively captures the synergies among various multi-omics data and comprehensively processes complex information. Through the feedback mechanisms of CycleGAN, CAEncoder learns to identify different distributions, resulting in improved generalization. The use of contrastive learning encourages the model to understand the relationships among different modalities, thereby enhancing data integration. The encoder maps high-dimensional data into a reduced latent space, where classification is subsequently performed. We evaluated the performance of the proposed model using various datasets, and the results demonstrate that it outperforms state-of-the-art methods. In the future, we plan to extend multi-omics data integration for biomarker detection and survival prediction using self-supervised learning. Furthermore, we recognize the importance of interpretability in deep learning models and plan to explore self-attention weight analysis and feature attribution methods to elucidate the contributions of different omics features to classification decisions, thereby enhancing the model's transparency and interpretability.

## Acknowledgments

**Financial Disclosure:** The authors extend their appreciation to Umm Al-Qura University, Saudi Arabia, for funding this research work through grant number: 25UQU4310136GSSR04. This research work was funded by Umm Al-Qura University, Saudi Arabia, under grant number: 25UQU4310136GSSR04.

## Author contributions

**Methodology:** Ma Yinghua, Ahmad Khan.

**Software:** Yang Heng.

**Writing – original draft:** Ma Yinghua.

**Writing – review & editing:** Ahmad Khan, Fiaz Gul Khan, Afnan Aldhahri, Iftikhar Ahmed Khan.

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
