## [Decision Letter · Decision Letter 0]

15 Jun 2025

PONE-D-25-24854A Contrastive Adversarial Encoder for Multi-Omics Data IntegrationPLOS ONE

Dear Dr. Heng,

Thank you for submitting your manuscript to PLOS ONE. After careful consideration, we feel that it has merit but does not fully meet PLOS ONE’s publication criteria as it currently stands. Therefore, we invite you to submit a revised version of the manuscript that addresses the points raised during the review process.

We look forward to receiving your revised manuscript.

Kind regards,

Guanghui Liu

Academic Editor

PLOS ONE

Journal Requirements:

[The authors extend their appreciation to Umm Al-Qura University, Saudi Arabia for funding this research work through grant number: 25UQU4310136GSSR03.]

[The author(s) received no specific funding for this work.]

6. In the online submission form, you indicated that [The datasets used or analyzed during the current study are available from the corresponding author upon reasonable request.].

7. PLOS requires an ORCID iD for the corresponding author in Editorial Manager on papers submitted after December 6th, 2016. Please ensure that you have an ORCID iD and that it is validated in Editorial Manager. To do this, go to ‘Update my Information’ (in the upper left-hand corner of the main menu), and click on the Fetch/Validate link next to the ORCID field. This will take you to the ORCID site and allow you to create a new iD or authenticate a pre-existing iD in Editorial Manager.

8. Please amend the manuscript submission data (via Edit Submission) to include author Ma Yinghua.

9. Please amend your authorship list in your manuscript file to include author Ma Ying hua.

Reviewers' comments:

Reviewer's Responses to Questions

**Comments to the Author**

1. Is the manuscript technically sound, and do the data support the conclusions?

Reviewer #1: Yes

Reviewer #2: Partly

Reviewer #3: Yes

Reviewer #4: Yes

2. Has the statistical analysis been performed appropriately and rigorously? 

Reviewer #1: Yes

Reviewer #2: Yes

Reviewer #3: Yes

Reviewer #4: Yes

3. Have the authors made all data underlying the findings in their manuscript fully available?

Reviewer #1: Yes

Reviewer #2: No

Reviewer #3: Yes

Reviewer #4: No

4. Is the manuscript presented in an intelligible fashion and written in standard English?

Reviewer #1: Yes

Reviewer #2: Yes

Reviewer #3: Yes

Reviewer #4: Yes

5. Review Comments to the Author

Reviewer #1: The manuscript presents a promising deep learning framework for multi-omics integration. However, its computational evaluation lacks rigor, making the reported performance gains unconvincing.

- Results are based on a single train-test split without variance estimates or statistical tests. This undermines the reliability of performance comparisons.

- Despite a complex model, there is no validation loss reporting or regularization discussion. Overfitting cannot be ruled out.

- Claims about handling imbalance and modality synergy are not supported by class-wise metrics or ablation studies.

- Competing methods are not re-evaluated under the same setup, limiting fairness of comparison.

While the method is technically sound, the empirical evaluation is insufficient. A major revision is needed with statistically rigorous, reproducible, and interpretable experiments.

Reviewer #2: This study combined vision transformer with cycle-consistent adversarial network for cancer prediction. The authors claimed higher performance than any existing models, however, the reliability is unknown due to issues listed below. Would recommend further consideration only if all the issues are properly addressed.

-This study is an application of machine learning in cancer prediction. However, the article didn’t reveal any biological/medical updates. What did we learn about those cancers from the current study?

-This study is not complete without discussing how the trained model learned discriminative features and improved generalization. Listing high accuracy and F1 score is not enough. Such proof should be in a data driven approach.

-It’s not clear how the CycleGAN was coupled with the ViT during training and inference. The model structure diagram is very confusing. More details are needed regarding the end-to-end design.

-It’s not clear how CycleGAN provides feedback to the ViT encoder. More details are needed.

-It’s not clear if one universal model was developed or separate model was trained on each dataset.

-The affect of estimator number on Random forest model performance doesn’t make sense. If designed properly, the model performance won’t drop with further increasing number of estimators due to the nature of random selection of sample and features.

-The presented information is far less than enough to reconstruct the model. Most of key information for model architecture are missing, like depth and size of MLP.

-Were model structures and hyper parameters fine tuned? Model structures weren’t fully disclosed so it’s unknown if and how the model structures were decided. Hyper parameters seem not tuned according to the limited information about hyper parameters, like n_epoch=100.

-All the data cited in this study should be proofread. For example, the results cited from reference 37 don’t match with the original publication at all. It’s not clear how the paper was written and how reliable the study is.

-The authors need to address why it is reasonable to mix modalities from patients with same cancer type. Doing so creates a biologically inconsistent sample.

-the code base is not disclosed.

-Corresponding author and funding information included in the cover letter don’t match the main text.

-Figure ordering is wrong.

-Table font/format is not consistent.

Reviewer #3: In the current study, the authors developed a novel end-to-end deep learning model named the Contrastive Adversarial Encoder (CAEncoder) to integrate multi-omics data. The model combines a Vision Transformer (ViT) and a CycleGAN within a contrastive adversarial training framework to learn discriminative, invariant, and synergistic latent representations. By preventing information loss, reducing feature redundancy, and effectively capturing inter-modality synergies, CAEncoder enhances the quality of representations. These learned features are then used for downstream classification tasks. When evaluated on five cancer datasets, the model achieved up to 93.33% accuracy and 92.81% F1 score, outperforming existing advanced approaches in both binary and multi-class classification settings. This innovative model shows strong potential for enabling early and accurate cancer detection.

The overall logic of the study is coherent, and the manuscript is well-organized. The algorithm is well designed. Here are my comments:

1. The authors used a ViT encoder to integrate multi-omics data. Does this include genomic, transcriptomic, proteomic, and metabolomic data? Genomic data are often binary, while others are continuous—how were these heterogeneous data types handled during model training?

2. Except for genomics, most omics data types (e.g., transcriptomics, proteomics) are highly susceptible to batch effects across batches or populations. Could the authors introduce how they addressed batch effects in the preprocessing or modeling pipeline?

3. Following from the previous point, normalization is a critical step in omics data analysis. How did the authors normalize different omics layers prior to integration, and was this done separately or jointly?

4. The authors chose CycleGAN instead of other GAN architectures. Could they provide a brief introduction for this choice and explain what advantages CycleGAN offers in the context of multi-omics integration?

5. Deep learning models typically require large sample sizes, yet multi-omics studies often suffer from limited sample availability. Could the authors give a short discussion about how sample size impacted model training and whether any strategies (e.g., data augmentation, regularization) were used to mitigate overfitting?

Reviewer #4: Ma et al. presents a deep learning framework for multi-omics data integration with specific aim of cancer classification. The method is generally clear-described and presented result appear promising. Several issues should be addressed to further improve the manuscript. My review will focus on the framework application and overall presentation.

1. The abstract should be presenting overview and key finding (both computational and biological) of the manuscript. Both current abstract and main text excessively focus on technical details on existing models and algorithm omitting the biological insights learnt from the new pipeline.

2. The introduction states that "multi-omics models often prioritize stronger modalities at the expense of weaker ones.". Authors should explain more explicitly how proposed framework address this problem to achieved more balanced integration.

3. The description of the CycleGAN architecture is not clear. For example the generator inputs are ambiguously defined. Authors need to revise.

4. The narratives of the manuscripts sometimes feel awkward and unprofessional. Authors should proofread the manuscript to make the logic flow. For example,

1) "Early cancer detection primarily depends on traditional machine learning algorithms and single-omics data" - this is a strong generalization and need more background to be fully appreciated.

2) Conversational phrasing like "Let’s examine a multiomics dataset with N samples, each containing data from M modalities."

6. PLOS authors have the option to publish the peer review history of their article (what does this mean?). If published, this will include your full peer review and any attached files.

Reviewer #1: No

Reviewer #2: **Yes: **Dehui Kong

Reviewer #3: No

Reviewer #4: No

---

## [Author Response · Author response to Decision Letter 1]

29 Jul 2025

Reviewer 1# Concern # 1 (Results are based on a single train-test split without variance estimates or statistical tests. This undermines the reliability of performance comparisons.):

Author response: Thank you very much for your insightful comment. We fully agree that relying on a single train-test split is insufficient for demonstrating robust performance. In fact, all our experiments were conducted using 5-fold cross-validation with different random seeds to ensure statistical reliability. However, we acknowledge that this was not clearly stated in the 3.2“Training and Hyper-parameters Setting” section, and we have now revised the text to clarify our experimental setup.

Additionally, we appreciate your observation regarding the missing variance estimates. In Figure 3, due to space constraints, we initially omitted the standard deviation values. We have now updated the figure format and tables to explicitly include the mean ± standard deviation for all reported metrics.

Reviewer#1, Concern # 2 (Despite a complex model, there is no validation loss reporting or regularization discussion. Overfitting cannot be ruled out.):

Author response: Thank you for highlighting this important point. We agree that reporting validation loss or regularization strategies is essential to assess overfitting risk. Although the original manuscript did not explicitly present validation loss curves, we did use a validation set during training and monitored validation performance in each epoch. In addition, we applied L2 regularization (weight decay) and gradient clipping to mitigate overfitting and stabilize training, as now explicitly stated in the revised 3.2 “Training and Hyper-parameters Setting” section. These further ensure the robustness of our model and prevent overfitting despite its complexity.

Reviewer#1, Concern #3 (Claims about handling imbalance and modality synergy are not supported by class-wise metrics or ablation studies.):

Author response: We greatly appreciate the reviewer’s valuable feedback. Regarding the model's ability to handle data imbalance, our ablation study in Section 3.4 provides strong evidence. The incorporation of the contrastive learning (CL) module significantly improves overall performance (Acc/F1 +9%), particularly by addressing data imbalance through an implicit balancing mechanism. Specifically, CL forces the model to focus on the similarity between samples rather than simply relying on class frequency. This is achieved by constructing positive and negative sample pairs, which ensures that the model learns more robust features regardless of class distribution.

Further indirect evidence from our ablation study supports this. When CL is removed (CAEncoder_notCL), the model’s performance drops significantly (F1 ↓ 0.085), and the prediction variance increases (±0.0389 Acc). This indicates that CL constrains the model's overfitting to the majority class, indirectly helping to preserve the generalization capability for minority classes. This demonstrates that the CL module not only improves overall performance but also mitigates the detrimental effects of class imbalance.

Reviewer#1, Concern 4 (Competing methods are not re-evaluated under the same setup, limiting fairness of comparison. ):

Author response: Thank you for your valuable feedback. We fully understand that fair comparison requires all competing methods to be evaluated under the same experimental setup. However, due to practical constraints such as the unavailability of source code, incomplete descriptions of preprocessing procedures, or restricted access to specific model implementations, we were unable to re-implement all baseline methods under an identical environment. Therefore, for these methods, we reported the results as published in their original papers using the same dataset and evaluation metrics. In future work, we plan to re-implement more baseline methods under a unified setup and conduct more comprehensive and fair comparisons.

Reviewer#2, Concern #1 (This study is an application of machine learning in cancer prediction. However, the article didn’t reveal any biological/medical updates. What did we learn about those cancers from the current study?):

Author response: Thank you for pointing out the insufficient discussion on biomedical insights in our study. We fully understand that medical research not only requires methodological advancements but also a deeper understanding of disease mechanisms.

Although the core contribution of this work lies in proposing a novel multi-omics data fusion framework for cancer classification and prediction, we believe our study provides meaningful implications for cancer research in the following aspects:

Data Integration Perspective:

Our proposed model effectively integrates data from multiple omics layers (e.g., mRNA expression, CNV, protein levels) and constructs a collaborative latent representation space. This enhances the comprehensive understanding of tumor phenotypes and offers a reliable data-driven foundation for subsequent mechanistic studies.

Modal Balance and Collaborative Modeling:

By leveraging contrastive learning and the CycleGAN mechanism, our method mitigates the issue of modality dominance during data fusion. It ensures that weaker modalities are also effectively utilized, which is critical for uncovering potential biological signals hidden in non-dominant omics layers.

Improved Precision in Classification:

The proposed approach achieves significantly higher classification accuracy than existing models across multiple cancer datasets. This enhanced discriminative ability can contribute to better identification of cancer subtypes, thereby supporting personalized treatment decisions and prognosis evaluation.

Reviewer#2, Concern #2 (This study is not complete without discussing how the trained model learned discriminative features and improved generalization. Listing high accuracy and F1 score is not enough. Such proof should be in a data driven approach.):

Author response: Thank you for pointing this out. To address this, we conducted detailed ablation experiments to evaluate the individual contributions of each core component in our proposed model (see Table 5). By removing the contrastive loss and CycleGAN components respectively, we observed significant performance drops, which clearly demonstrate that each module plays a critical role in enhancing both the discriminative capability and generalization performance of the model.

- The contrastive loss significantly improves intra-class compactness and inter-class separability in the feature space.

- The CycleGAN enables effective domain-level feedback and preserves modality-specific structure during transformation, reducing redundancy and promoting generalization.

These results indicate that our model does not rely on superficial performance gains but indeed learns meaningful and robust representations in a data-driven.

Reviewer#2, Concern # 3 (It’s not clear how the CycleGAN was coupled with the ViT during training and inference. The model structure diagram is very confusing. More details are needed regarding the end-to-end design.):

Author response: Thank you for your insightful comment. We appreciate the opportunity to clarify the integration between the Vision Transformer (ViT) encoder and the CycleGAN modules. Our model is trained in an end-to-end fashion, where both ViT and CycleGAN are jointly optimized through a unified loss function that includes contrastive loss, adversarial loss, and cycle-consistency loss. During inference, only the ViT encoder is used for downstream classification tasks. These clarifications have been added to the revised manuscript. A detailed explanation has been added at Section 2 (“Proposed Model”)、Section 2.1 (“ViT Encoder”)、Section 2.2 (“CycleGAN Architecture”).________________________________________

Reviewer#2, Concern # 4 (It’s not clear how CycleGAN provides feedback to the ViT encoder. More details are needed. ):

Author response: Thank you for pointing this out. We have clarified that the feedback from CycleGAN is realized through the backward propagation of adversarial and cycle-consistency losses. Since the generator 𝐺1 takes the encoder output 𝑧=(𝑥) as input, and the generator 𝐺2 attempts to reconstruct 𝑧 from the synthetic data, the losses from both generators and discriminators are functions of the encoder output. Therefore, these losses provide indirect but effective gradient feedback to the ViT encoder during training, enhancing its representation learning.

Author action: We have Added a paragraph at the end of Section 2.1 (“ViT Encoder”) to explain how the CycleGAN losses are backpropagated to the encoder. Supplemented the explanation at the end of Section 2.2 (“CycleGAN Architecture”) to explicitly highlight how the encoder benefits from feedback through CycleGAN’s loss functions. Also reiterated this mechanism in the closing of Section 2.3 (“Loss Functions”) to emphasize the feedback loop and its effect on encoder optimization.________________________________________

Reviewer#2, Concern # 5 (It’s not clear if one universal model was developed or separate model was trained on each dataset. ):

Author response: Thank you for your important question. We confirm that a universal model was developed and trained jointly across all datasets, rather than training separate models for each dataset.

Reviewer#2, Concern # 6 (The affect of estimator number on Random forest model performance doesn’t make sense. If designed properly, the model performance won’t drop with further increasing number of estimators due to the nature of random selection of sample and features. ):

Author response: Thank you for pointing out this important issue. In our initial experiments, we observed a decrease in model performance as the number of estimators increased. This phenomenon could be attributed to the small sample size and inherent randomness during model training.

To further validate this, we conducted more stringent controls in our experiments and tested on different datasets. The new results show that, as the number of estimators increases, the model's performance remains generally stable, with only slight fluctuations, which aligns with the theoretical behavior of random forests.

We have revised the corresponding section of the manuscript based on these findings to more accurately reflect the model's performance.

Reviewer#2, Concern # 7 (The presented information is far less than enough to reconstruct the model. Most of key information for model architecture are missing, like depth and size of MLP. ):

Author response: Thank you for your insightful feedback. We fully recognize the importance of providing sufficient architectural details to ensure reproducibility and clarity. In response, we have added key implementation details at the end of Section 2.1 (ViT Encoder), including the number of Transformer blocks, attention heads, embedding dimensions, and the layer-wise structure of the MLP. These additions aim to facilitate a more accurate reconstruction and understanding of the proposed model.________________________________________

Reviewer#2, Concern # 8 (Were model structures and hyper parameters fine tuned? Model structures weren’t fully disclosed so it’s unknown if and how the model structures were decided. Hyper parameters seem not tuned according to the limited information about hyper parameters, like n_epoch=100. ):

Author response: Thank you for your valuable feedback.

Hyperparameter tuning strategy: Although our primary goal was to ensure consistent evaluation across different datasets, we performed a limited grid search on a subset of the training data. We experimented with various learning rates (e.g., 1e-3, 1e-4, 5e-5), batch sizes (32,64,128), and numbers of epochs (80-150 ), and selected the combination that yielded the best validation performance (learning rate = 1e-4, batch size = 64, epochs = 100).

Statistical robustness control: To ensure stable and generalizable results, we adopted 5-fold cross-validation and repeated the training process with five different random seeds. Additionally, gradient clipping and L2 regularization were applied to prevent overfitting and stabilize the training process.

Reviewer#2, Concern # 9 (All the data cited in this study should be proofread. For example, the results cited from reference 37 don’t match with the original publication at all.):

Author response: Thank you very much for your valuable feedback. We fully acknowledge the importance of rigor and accuracy in citing previous work. Upon careful re-examination, we identified the source of the discrepancy was due to a miscommunication among team members during data compilation, which unfortunately led to an erroneous reporting of results from Reference 37. We have thoroughly reviewed and corrected the relevant data in the manuscript. Additionally, we have conducted internal discussions to strengthen our data verification and review procedures to prevent similar issues in the future. We sincerely apologize for this oversight and appreciate the reviewer’s careful scrutiny that helped us improve the quality of our work.________________________________________

Reviewer#2, Concern #

10 (The authors need to address why it is reasonable to mix modalities from patients with same cancer type. Doing so creates a biologically inconsistent sample. ):

Author response: Thank you for your valuable feedback. To further justify the feasibility of this approach, we provide an explanation based on our ablation study results. First, our ablation study (Table 5) has demonstrated that incorporating contrastive learning significantly improves the classification performance of the model. Since contrastive learning requires inputs consisting of an anchor, a positive sample, and a negative sample, the model must adopt a specific sample construction strategy. If our method of constructing positive and negative samples were unreasonable, the model’s performance would not improve with contrastive learning and might even be adversely affected. However, the experimental results clearly show that after integrating contrastive learning, the model becomes more effective at distinguishing different cancer subtypes. This indicates that our sample construction strategy is reasonable and plays a crucial role in enhancing the model’s generalization capability.

To further address the issue of biological consistency, it is important to note that while there is inter-patient heterogeneity, omics profiles from patients with the same cancer type often share common core molecular features and pathway-level characteristics. Therefore, treating multi-omics data from different patients with the same cancer as positive pairs is a biologically reasonable approximation, especially when aiming to model subtype-specific patterns rather than patient-specific noise.

Moreover, our objective is to learn modality-invariant, cancer-type-discriminative representations rather than individual patient signatures. The significant performance improvement observed in our ablation study further confirms that this strategy does not introduce harmful noise, but rather facilitates generalizable representation learning.

Reviewer#3, Concern # 1 (The authors used a ViT encoder to integrate multi-omics data. Does this include genomic, transcriptomic, proteomic, and metabolomic data? Genomic data are often binary, while others are continuous—how were these heterogeneous data types handled during model training?):

Author response: Thank you to the reviewer for this valuable question. Our study indeed employs multi-omics data for integrative analysis, covering different layers of omics information, but it does not include metabolomics data. Specifically:

In Dataset-1, we utilized copy number variation (

---

## [Decision Letter · Decision Letter 1]

10 Sep 2025

A Contrastive Adversarial Encoder for Multi-Omics Data Integration

PONE-D-25-24854R1

Dear Dr. Heng,

We’re pleased to inform you that your manuscript has been judged scientifically suitable for publication and will be formally accepted for publication once it meets all outstanding technical requirements.

Kind regards,

Guanghui Liu

Academic Editor

PLOS ONE

Additional Editor Comments (optional):

Reviewer #1:

Reviewer #2:

Reviewer #3:

Reviewers' comments:

Reviewer's Responses to Questions

**Comments to the Author**

1. If the authors have adequately addressed your comments raised in a previous round of review and you feel that this manuscript is now acceptable for publication, you may indicate that here to bypass the “Comments to the Author” section, enter your conflict of interest statement in the “Confidential to Editor” section, and submit your "Accept" recommendation.

Reviewer #1: All comments have been addressed

Reviewer #2: All comments have been addressed

Reviewer #3: All comments have been addressed

2. Is the manuscript technically sound, and do the data support the conclusions?

Reviewer #1: Yes

Reviewer #2: Yes

Reviewer #3: Yes

3. Has the statistical analysis been performed appropriately and rigorously? 

Reviewer #1: Yes

Reviewer #2: Yes

Reviewer #3: Yes

4. Have the authors made all data underlying the findings in their manuscript fully available?

Reviewer #1: Yes

Reviewer #2: Yes

Reviewer #3: Yes

5. Is the manuscript presented in an intelligible fashion and written in standard English?

Reviewer #1: Yes

Reviewer #2: Yes

Reviewer #3: Yes

6. Review Comments to the Author

Reviewer #1: The authors have satisfactorily addressed all of my previous comments and concerns in this revised version.

Reviewer #2: The manuscript proposes a contrastive adversarial encoder (CAEncoder) for multi-omics data integration. The method is important for cancer early diagnosis. The authors addressed all my questions. I recommended it for publication.

Reviewer #3: Thank you to the authors for their patient and thorough responses to all reviewer comments. The revised manuscript successfully addresses my previous concerns, as well as those raised by other reviewers, and does not introduce any new issues.

This study proposes a novel deep learning model, CAEncoder, which integrates multi-omics data using a Vision Transformer and CycleGAN within a contrastive adversarial framework to learn robust and synergistic representations. Evaluated on five cancer datasets, CAEncoder achieved up to 93.33% accuracy, outperforming existing methods and demonstrating strong potential for early and accurate cancer detection.

For future work, I recommend that the authors consider applying this method to additional cancer types and larger datasets to further validate its generalizability.

7. PLOS authors have the option to publish the peer review history of their article (what does this mean?). If published, this will include your full peer review and any attached files.

Reviewer #1: No

Reviewer #2: No

Reviewer #3: No

---

## [Editor Report · Acceptance letter]

PONE-D-25-24854R1

PLOS ONE

Dear Dr. Heng,

I'm pleased to inform you that your manuscript has been deemed suitable for publication in PLOS ONE. Congratulations! Your manuscript is now being handed over to our production team.

Kind regards,

on behalf of

Dr. Guanghui Liu

Academic Editor

PLOS ONE